# Parametric Synthesis of Single-Stage Lattice-Type Acoustic Wave Filters and Extended Multi-Stage Design

**DOI:** 10.3390/mi15091075

**Published:** 2024-08-26

**Authors:** Wei-Hsien Tseng, Ruey-Beei Wu

**Affiliations:** Graduate Institute of Communication Engineering, National Taiwan University, Taipei 10617, Taiwan; s10482013@gmail.com

**Keywords:** lattice topology, wideband, parametric analysis, acoustic filter, BVD model, Chebyshev response, optimization method

## Abstract

This study proposes a single-stage lattice-type acoustic filter using an analytical solution method for either a narrow passband filter or a wider passband filter using two kinds of parameter assignments in the Butterworth–Van Dyke (BVD) model. To achieve the goal of a large bandwidth or high return loss, two first-order all-pass conditions are used. For multi-stage lattice-type filters, the cost function is defined and design parameters are extracted by using pattern search, while the initial values are provided through single-stage design to shorten optimization time and allow convergence to a better solution. This method provides the S-parameter frequency response for the filter on the YX 42° cut angle of lithium tantalate (electromechanical coupling coefficient of about 6%) that can meet the system specifications as much as possible. Finally, the three-stage lattice-type was applied to various 5G bands with a fractional bandwidth of 2–5%, resulting in a passband return loss of 10 dB and an out-of-band rejection of 40 dB or more.

## 1. Introduction

Recent advancements in 5G wireless communication have expanded spectrum frequencies and bands, necessitating surface acoustic wave (SAW) filters capable of higher frequencies and larger bandwidths [1]. Traditional SAW materials like bulk LiNbO_3_ (LN) and LiTaO_3_ (LT) face challenges in meeting 5G bandwidth requirements, prompting innovations in thin-film technologies [2].

Experimental results using 32°Y-LN/SiO_2_/p-Si/Si structures have achieved SAW filters covering the 5G n77 and n78 bands with smooth passbands, showcasing advancements in filter design and fabrication techniques [3]. Efforts also focus on SAW heterostructures and third-generation semiconductors like SiC to enhance filter performance and scalability for future 5G applications [4]. Therefore, it is necessary to study wideband filters [5].

At present, the vast majority of mobile phone RF filters are acoustic filters based on surface acoustic waves (SAWs) and body acoustic waves (BAWs). They are popular due to the best combination of excellent performance (high quality factor, low insertion loss, high out-of-band rejection, etc.) with miniaturization and low cost [6].

The bandwidth of an acoustic filter is restricted by its electromechanical coupling coefficient. Previous research has explored different filter synthesis methods under the assumption of unchanged piezoelectric material properties [7]. Ladder- and lattice-type configurations are the most prevalent combinations, each with their own unique strengths and limitations. Although the traditional ladder-type structure is commonly used for combining filters with broader bandwidth, it often fails to achieve optimal wideband performance [8]. A modified lattice configuration comprising auxiliary shunt inductors is proposed to achieve a wideband filter response using AlN-based BAW resonators [9]. Although it exhibits larger bandwidth, its insertion loss is degraded as a tradeoff.

Resonators with the same electromechanical coupling coefficient can use different filter combination architectures to achieve different fractional bandwidths. The use of lattice-type filter combinations can achieve the highest fractional bandwidth of the filter with the same electromechanical coupling coefficient [10]. Therefore, we will develop a filter synthesis method based on the lattice combination method to meet 5G’s demand for high bandwidth filters in the Sub-6G band.

The remainder of this study is organized as follows. Section 2.1 discusses the lossless BVD model for the acoustic resonators assumed in the design. Network theory is used to analyze a lattice network, which is then employed to synthesize narrow-band Chebyshev filter design. Section 2.3 addresses the performance limitations of the synthesized narrow-band filters and proposes a new design to yield a wide-passband design for single-stage lattice filters. Section 2.5 extends the design to multi-stage lattice filters to improve filter performance, especially in band selectivity and out-of-band rejection. This method is then applied to design filters for various 5G bands. Brief conclusions are given in Section 3.

## 2. Basic Synthesis Theory of Single-Stage Filter

### 2.1. Equivalent Circuit of Acoustic Resonator

In terms of acoustic resonators, the lossless BVD (Butterworth–Van Dyke) model is simple and convenient to model for parameter extraction and design [11]. The equivalent circuit diagram is shown in Figure 1.

To reduce the time for parametric analysis and design optimization, the number of variables must be reduced. Since the S-parameter variables are the circuit parameters $L_{m}{,C}_{m},C_{o}$ for the BVD model, new variables are defined: the impedance $Z_{m}=\sqrt{L_{m}/C_{m}}$, the resonant frequency $f_{r}=1/2\pi \sqrt{L_{m}C_{m}}$, and the anti-resonant frequency $f_{a}=f_{r}\cdot \sqrt{1+C_{m}/C_{0}}$ for the resonator.

Piezoelectric materials have a different electromechanical coupling coefficient, $k_{t}^{2}$, which is defined as the ratio of the total input electrical energy to the generated acoustic energy and is used to indicate the strength of piezoelectric conversion characteristics. $k_{t}^{2}$ can be expressed in terms of $f_{r}$ and $f_{a}$ as follows:(1)$$k_{t}^{2}=1-{\left(\frac{f_{r}}{f_{a}}\right)}^{2}=\frac{C_{m}}{C_{m}+C_{0}}$$

For a specific electromechanical coupling coefficient $k_{t}^{2}$, $Z_{m}$ and $f_{r}$ are used to replace $L_{m},C_{m},C_{o}$, and the variables can be reduced from 3 to 2. The conversion relationship is $L_{m}=Z_{m}/{2\pi f}_{r},\;C_{m}=1/{2\pi Z}_{m}f_{r},\;and\;C_{o}=C_{m}\cdot \left(1-k_{t}^{2}\right)/k_{t}^{2}$. The s-domain impedance for the resonator using the BVD model shown in Figure 1 is as follows:(2)$$Z_{BVD}=\frac{1}{s}\cdot \frac{{Z_{m}k_{t}^{2}\omega_{r}(s}^{2}+\omega_{r}^{2})}{s^{2}\left(k_{t}^{2}-1\right)+\omega_{r}^{2}}\;$$
where $\omega_{r}=2\pi f_{r}$.

Acoustic resonators typically exhibit resonance and anti-resonance. Near the resonant frequency $f_{r}$, the resonator behaves like a series LC resonant circuit with an effective inductance $L_{m}$ and capacitance $C_{m}$, as shown in Figure 1.

Near the anti-resonant frequency $f_{a}$, the resonator behaves like a shunt LC resonant circuit. The effective inductance and capacitance are derived using the equivalence of admittance and are, respectively, defined as follows:(3)$$L_{eff}={L_{m}\left(k_{t}^{2}\right)}^{2}\;\mathrm{and}\;C_{eff}=C_{m}\frac{{1-k}_{t}^{2}}{{\left(k_{t}^{2}\right)}^{2}}$$

For a specific acoustic material with a specific value for $k_{t}^{2},\;$the set of variables for the S-parameters for the lattice filter is the set of all BVD model parameters:(4)$$\;\;\vec{x}=\left[Z_{m1},Z_{m2}{{,f}_{r1},f}_{r2}\right]$$

### 2.2. One Stage Lattice Network Analysis

To derive the *S*-parameters for the single-stage lattice network shown in Figure 2, we first consider the *Z* matrix of the network. By assuming open circuit at Port 2, i.e., *I*_2_ = 0, the voltages at Port 1 and Port 2 can be given by $v_{1}=\frac{1}{2}\left(Z_{s}+Z_{p}\right)\cdot i_{1}$ and $v_{2}=\frac{Z_{p}-Z_{s}}{Z_{p}+Z_{s}}\cdot v_{1}$. As a result, $Z_{11}=v_{1}/i_{1}=\frac{1}{2}\left(Z_{s}+Z_{p}\right)$, $Z_{21}=v_{2}/i_{1}=\frac{1}{2}\left(Z_{p}-Z_{s}\right)$. The network is symmetric, so the *Z* matrix is written as follows: (5)$$\left[Z\right]=\frac{1}{2}\cdot \left[\begin{array}{cc} Z_{p}+Z_{s} & Z_{p}-Z_{s}\\ Z_{p}-Z_{s} & Z_{p}+Z_{s} \end{array}\right]\;\;$$

Converting from *Z*-parameters to *S*-parameters yields the following [12]:(6)$$\left[S\right]=\left[\begin{array}{cc} \frac{Z_{p}Z_{s}-{Z_{0}}^{2}}{\left(Z_{p}+Z_{0}\right)\left(Z_{s}+Z_{0}\right)} & \frac{\left(Z_{p}-Z_{s}\right)Z_{0}}{\left(Z_{p}+Z_{0}\right)\left(Z_{s}+Z_{0}\right)}\\ \frac{\left(Z_{p}-Z_{s}\right)Z_{0}}{\left(Z_{p}+Z_{0}\right)\left(Z_{s}+Z_{0}\right)} & \frac{Z_{p}Z_{s}-{Z_{0}}^{2}}{\left(Z_{p}+Z_{0}\right)\left(Z_{s}+Z_{0}\right)} \end{array}\right]\;$$

### 2.3. Synthesis for a Narrow-Band Single-Stage Design

Using Analytic Expression (6), a single-stage lattice network synthesizes a second-order Chebyshev bandpass filter. It is well known that the *S*_12_ of the second-order Chebyshev bandpass filter has four poles. On the other hand, both $Z_{p}$ and $Z_{s}$ behave as series and shunt LC resonant circuits near resonance or anti-resonance, respectively. Therefore, each set of $Z_{p}+Z_{0}$ and $Z_{s}+Z_{0}$ has two zeros, which together contribute to the four poles of *S*_12_ in (6). The design is obtained by equating the four poles with those in the second-order Chebyshev filter.

Select the design example of a second-order Chebyshev bandpass filter with the following specifications: $L_{Ar}$= 0.4576 dB or return loss (RL) = 10 dB; FBW =1%; $f_{o}=2\;GHz$; and $Z_{o}=50\;\Omega$ using a lattice filter in the acoustic material of lithium tantalate with $k_{t}^{2}=6.1\%$. 

If series LC resonators are used to synthesize the filter, the BVD model parameters are used to determine $L_{m}=543.94\;and\;538.42\;nH;\;C_{m}=11.761\;and\;11.642\;fF$; and $C_{o}=181.05\;and\;179.21\;fF,$ i.e., $\vec{x}=$ [6800.6, 6800.6; 1.9898$\times 10^{9}$, 2.0102$\times 10^{9}$]. Figure 3 shows the final S-parameters and the constituent impedances $Z_{BVD}$ versus frequency. It was validated that the design successfully met the specification requirements. 

If shunt LC resonators are used to synthesize filters, the effective inductance $L_{eff}$ and capacitance $C_{eff}$ are determined. Then, use (3) to obtain the BVD model parameters: $L_{m}=$ 7.9020 and 7.8218 nH; $C_{m}=$ 0.86220 and 0.85345 pF; and $C_{o}$ = 13.272 and 13.138 pF, i.e., $\vec{x}=$[95.7333, 95.7333; 1.9282$\times 10^{9}$, 1.9480$\times 10^{9}$]. Figure 4 shows the final *S*-parameters and the constituent impedances $Z_{BVD}$ versus frequency. It was validated that the design successfully met the specification requirements.

### 2.4. Passband Conditions

Most acoustic materials have small $k_{t}^{2}$, and the lattice filter synthesized by the previous approach has a narrow passband. This section investigates the conditions under which lattice networks form passbands and out-of-band rejection. 

The lattice network can be all-pass if $S_{11}=0$ in (6), i.e.,
(7)$$Z_{p}Z_{s}=Z_{0}^{2}$$

A possible solution is that $Z_{p}$ is inductive and $Z_{s}$ capacitive, or vice versa, in the frequency range of interest [9].

When it is impossible to synthesize a wide passband filter with a one-order all-pass filter, two one-order all-pass filters are combined to achieve filters with a wider passband. A fruitful strategy is two acoustic resonators in one all-passband, where one is inductive and one is capacitive; in the other all-passband, one is capacitive and the other is inductive. The passband becomes wider because the all-pass conditions occur in two bands.

The relationship between the BVD model parameters for the two resonators in the middle figure of Figure 5 is determined. In band 1 from $f_{r1}$ to $f_{a1}$, $Z_{s}$ is inductive and near anti-resonance while $Z_{p}$ is capacitive and near resonance. As a result, (7) gives $\sqrt{L_{eff,1}/C_{m,2}}=Z_{0}$ and by (3), it yields
(8)$$\sqrt{\frac{L_{m,1}}{C_{m,2}}}≅\frac{Z_{0}}{k_{t}^{2}}\;\;$$

In band 2 from $f_{r2}$ to $f_{a2}$, $Z_{p}$ is inductive and near resonance while $Z_{s}$ is capacitive and near anti-resonance. As a result, (7) gives $\sqrt{L_{m,2}/C_{eff,1}}=Z_{0}$ and by (3), it yields
(9)$$\sqrt{\frac{L_{m,2}}{C_{m,1}}}≅\frac{\sqrt{1-k_{t}^{2}}}{k_{t}^{2}}Z_{0}$$

To simplify the variables, we replace $L_{m},C_{m},C_{o}$ with $Z_{m}$ and $f_{r}.$ From (8) and (9),
(10)$$\sqrt{Z_{m,1}\cdot Z_{m,2}}=\sqrt[{4}]{\frac{L_{m,1}}{C_{m,2}}\cdot \frac{L_{m,2}}{C_{m,1}}}≅\frac{Z_{0}}{k_{t}^{2}}\cdot \sqrt[{4}]{1-k_{t}^{2}}$$

Since $k_{t}^{2}$ is much smaller than 1, the relationship between the two passbands is simplified as follows:(11)$$\sqrt{Z_{m,1}{\cdot Z}_{m,2}}≅\frac{Z_{0}}{k_{t}^{2}}$$

For $Z_{0}=50$; $k_{t}^{2}=0.061$, two examples are used to illustrate this mechanism. The first example is ${\vec{x}}_{1}=$ [1011, 664.3; 0.966141, 1.000351], where the specification is FBW = 1%; RL = 16 dB. The plots for the S-parameter versus frequency and the imaginary part of the acoustic resonator impedance are shown in Figure 5. The second example is ${\vec{x}}_{2}=$ [1017, 660.8; 0.9614, 1.00849], which is specified as FBW = 5%; RL = 9.6 dB. The frequency response of the S-parameter and the imaginary part of the acoustic resonator impedance are shown in Figure 6. 

These two examples show that there are two full passbands, the resonator impedances are both pairwise (one inductive and the other capacitive), and the geometric mean for the two impedances should be around $Z_{0}/k_{t}^{2}$. When we want to design a wide bandwidth, we let the two resonators $f_{a1}$ and $f_{r2}$ become far away from each other, and the resonator impedances between $f_{a1}$ and $f_{r2}$ are no longer pairwise. As for how wide the bandwidth can be in this method, it is to let the two passbands slowly separate. The greater the separation, the smaller the RL value is.

### 2.5. Multi-Stage Design

Now that the single-stage lattice filter can have a wider bandwidth by the present design, its selectivity and out-of-band rejection are not sufficient. Increasing the number of stages can provide a better solution, but the number of variables becomes larger. Nonetheless, we can use the solution for a single-stage lattice filter as the initial value to shorten the optimization time.

### 2.6. Mult-Stage Lattice Network Analysis

The ABCD matrix is commonly used to solve the multi-stage lattice filters shown in Figure 7. The *Z* matrix for the single-stage lattice network in (5) is converted into an ABCD matrix, i.e.,
(12)$${\left[\begin{array}{cc} A & B\\ C & D \end{array}\right]}_{1-stage}=\frac{1}{Z_{21}}\cdot \left[\begin{array}{cc} Z_{11} & Z_{11}Z_{22}-Z_{12}Z_{21}\\ 1 & Z_{22} \end{array}\right]$$

The ABCD matrix of the n-stage lattice filter can be obtained by multiplying the ABCD matrix of single-stage lattice filters,
(13)$${\left[\begin{array}{cc} A & B\\ C & D \end{array}\right]}_{n-stage}={\left[\begin{array}{cc} A & B\\ C & D \end{array}\right]}_{1st-stage}\cdots {\left[\begin{array}{cc} A & B\\ C & D \end{array}\right]}_{n’th-stage}$$

Finally, using the ABCD matrix and S-parameter conversion formula [12], the S-parameters of the *n*-stage lattice filter are as follows:(14)$${\left[S\right]}_{n-stage\;}=\left[\begin{array}{cc} \frac{A+B/Z_{0}-CZ_{0}-D}{A+B/Z_{0}+CZ_{0}+D} & \frac{2\left(AD-BC\right)}{A+B/Z_{0}+CZ_{0}+D}\\ \frac{2}{A+B/Z_{0}+CZ_{0}+D} & \frac{-A+B/Z_{0}-CZ_{0}+D}{A+B/Z_{0}+CZ_{0}+D} \end{array}\right]$$

The set of S-parameters of the multi-stage lattice filter is the set of all BVD model parameters, i.e.,
(15)$$\vec{x}=\left[Z_{m1\;},\;Z_{m2\;}{{,\ldots ,Z_{mn},Z_{m\left(n+1\right)};f}_{r1\;},f}_{r2},\ldots ,f_{rn},f_{r(n+1)}\right]$$

### 2.7. Pole-Zero Distribution Optimization Process

For the design of multi-stage lattice filters, the concept is to optimize the pole-zero distribution so that the positions of the poles and zeros of $S_{11}$ of the lattice network in Figure 7 are as close as possible to those of the target filter.

First, find the pole-zero distribution of the target response. In this study, we choose the Chebyshev bandpass filters as design examples, although other bandpass filters can be used as well. We define a cost function and optimize it using a pattern search method [13] while setting appropriate initial values. The root-mean-square error (RMSE) between the target pole-zero distribution and that of the lattice filter is calculated in each iteration until the cost function converges. The flow chart is shown in Figure 8.

The pole-zero extraction for $S_{11}$ in the lattice network in (14) uses the MATLAB R2020a numerical methods ***numden*** and ***sym2poly***. The constants are organized into rational function form, and $S_{11}$ is written as follows:(16)$$S_{11}=K\frac{\left(s-z_{r1})(s+z_{r2}\right)}{\left(s+p_{r1})(s+p_{r2}\right)}\frac{\prod_{j=1}^{2n}\left(s-z_{j})(s-z_{j}^{*}\right)}{\prod_{j=1}^{2n}\left(s-p_{j})(s-p_{j}^{*}\right)}$$

For a lossless n-stage lattice filter with 4*n* acoustic resonators, the poles and zeros in (16) can be divided into two parts. In the first part, $z_{r1}$ and $z_{r2}$ are associated with zeros located on the real axis, and $p_{r1}$ and $p_{r2}$ are associated with poles located on the real axis. In the second part, there are 2*n* conjugate pairs of complex zeros and 2*n* conjugate pairs of complex poles. These complex zeros and poles dominate the in-band response, so only they are optimized using the following method.

The cost function is defined as the RMSE between the target pole-zero and the pole-zero of the lattice filter divided by the fractional bandwidth of the target filter, i.e.,
(17)$$\mathrm{cost}=\frac{1}{FBW}\cdot \sqrt{\sum_{j=1}^{2n}\left\{{\left|p_{j,\mathrm{tar}.}-p_{j}\left(\vec{x}\right)\right|}^{2}+{\left|z_{j,\mathrm{tar}.}-z_{j}\left(\vec{x}\right)\right|}^{2}\right\}}$$

The following is an example of a three-stage lattice design using pole-zero distribution optimization. The target setting is an FBW = 5.1% and an RL = 11 dB for a sixth-order Chebyshev response. The material for the acoustic resonator is lithium tantalite with a YX 42° cut angle and $k_{t}^{2}=0.061$. To facilitate optimization, a new variable $\delta =\frac{Z_{m2}}{Z_{m1}}$ is defined. If the value of δ is known, and given the relation in (10), we can obtain $Z_{m,1}\;$and $Z_{m2}$.

Our initial values are set to δ=2 and $f_{r2}/f_{r1}=1.05$ for the first stage, $\delta =0.25\;and\;f_{r2}/f_{r1}=1.05$ for the second stage, and $\delta =2\;and\;f_{r2}/f_{r1}=1.05$ for the third stage. The pole-zero distribution of the target and initial S-parameter is shown in Figure 9, where circles and squares are zeros, x marks and stars are poles, red is the target, and black is the lattice filter S-parameter. Then, the cost function (17) is optimized to minimize the distance between the pole-zero distribution and the target. As shown in Figure 10, the cost function converges versus the iteration number during the pole-zero distribution optimization. The optimized S-parameters for the three-stage lattice filter are shown in Figure 11.

The normalized frequency response of the S-parameters for an optimized three-stage lattice filter is shown in Figure 12. The proposed design achieves an FBW = 5.06% and RL = 9.7 dB in the passband. The BVD model parameters of the acoustic resonators in the lattice filter are listed in Table 1. A performance comparison between second-order and sixth-order lattice-type filters with the same bandwidth is shown in Figure 13. It is verified that the multi-stage lattice filter design greatly improves selectivity and out-of-band rejection. 

### 2.8. Conversion Formula of the New System Parameters

If the center frequency of the design is $f_{o}^{\prime }$ and the characteristic impedance is $Z_{o}^{\prime }$, the BVD parameters obtained from the previous design examples need to be converted. The conversion equations are as follows:(18)$$Z_{m}^{\prime }=Z_{m}Z_{o}^{\prime };f_{r}^{\prime }=f_{r}f_{o}^{\prime }$$

Or
(19)$$L_{m}^{\prime }=L_{m}\frac{Z_{o}^{\prime }}{f_{o}^{\prime }};C_{m}^{\prime }=C_{m}\frac{1}{{Z_{o}^{\prime }f}_{o}^{\prime }};C_{o}^{\prime }=C_{m}^{\prime }\frac{1-k_{t}^{2}}{k_{t}^{2}}$$

### 2.9. Application in the 5G Bands

The three-stage lattice filter, utilizing the new system conversion formula, is used to design the 5G bands, as shown in Table 2. For an acoustic material with a value of $k_{t}^{2}=0.061$, the 5G bands with an FBW value of less than 5% are designed using the methods of this study. The design meets the unified requirements for RL greater than 10 dB. 

Most 5G bands have a bandwidth range of about 1.9–5.1%. These bands can be accommodated through the present design and are highlighted in Table 2. Figure 14 shows the frequency response of the S-parameters of these examples in the first phase of design when the acoustic resonators are assumed to be lossless. The implementation of acoustic resonators and typical considerations in package layout design have been described for filters [14], multiplexers [15], and antennaplexers [16], but they are omitted in this study.

## 3. Conclusions

This study proposes a design flow for lattice-type acoustic filters, where a single-stage lattice provides an accurate solution to directly determine the BVD model parameters. Multi-stage lattice-type filters use the results of the single-stage design to establish the initial values for optimized performance.

A three-stage lattice design is used for 5G bands with a fractional bandwidth of 2–5%, providing a passband return loss of 10 dB and an out-of-band rejection of more than 40 dB.

## Figures and Tables

**Figure 1 micromachines-15-01075-f001:**
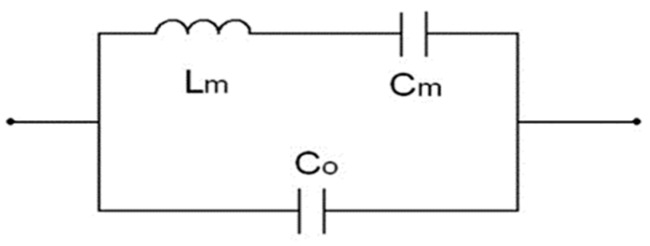
Lossless BVD model for acoustic wave resonators.

**Figure 2 micromachines-15-01075-f002:**
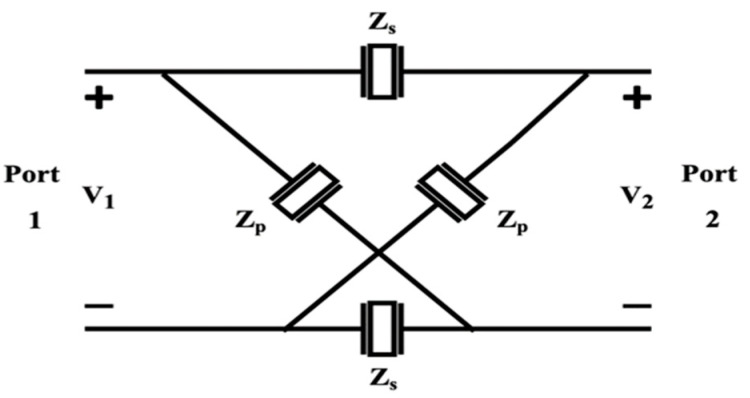
Topology of a single-stage lattice network.

**Figure 3 micromachines-15-01075-f003:**
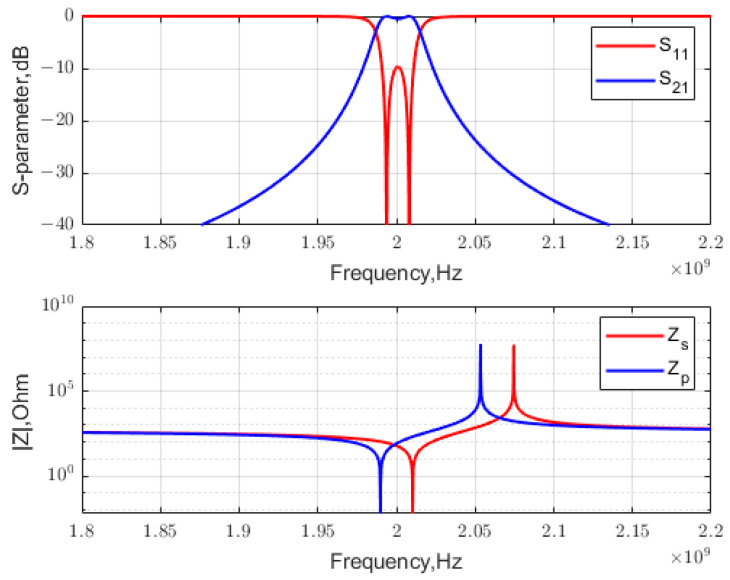
The lattice filter synthesized near resonance.

**Figure 4 micromachines-15-01075-f004:**
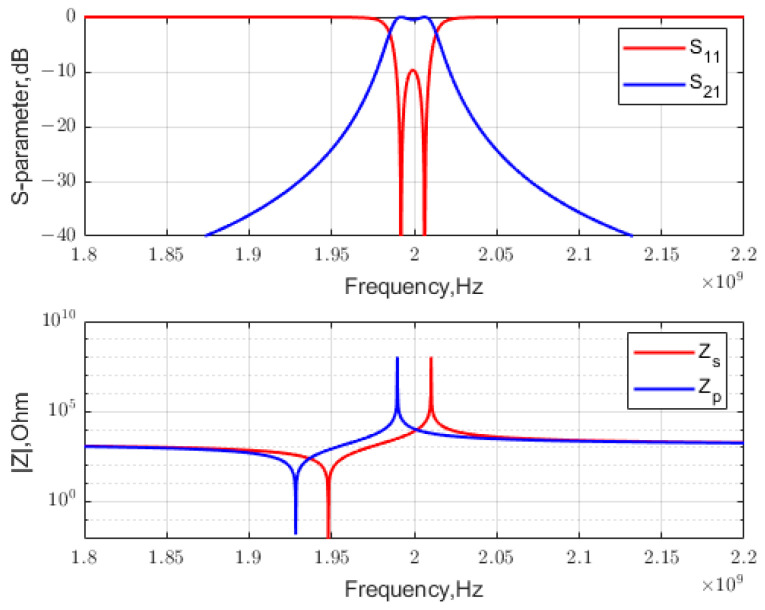
The lattice filter synthesized near anti-resonance.

**Figure 5 micromachines-15-01075-f005:**
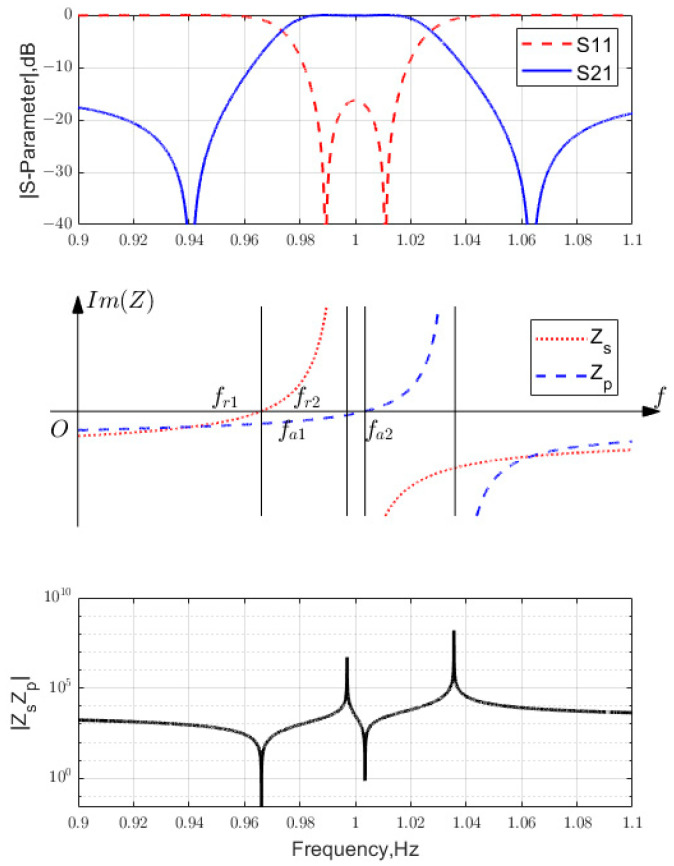
Frequency response of S-parameter and the imaginary part of the acoustic resonator impedance (variable is ${\vec{x}}_{1}$).

**Figure 6 micromachines-15-01075-f006:**
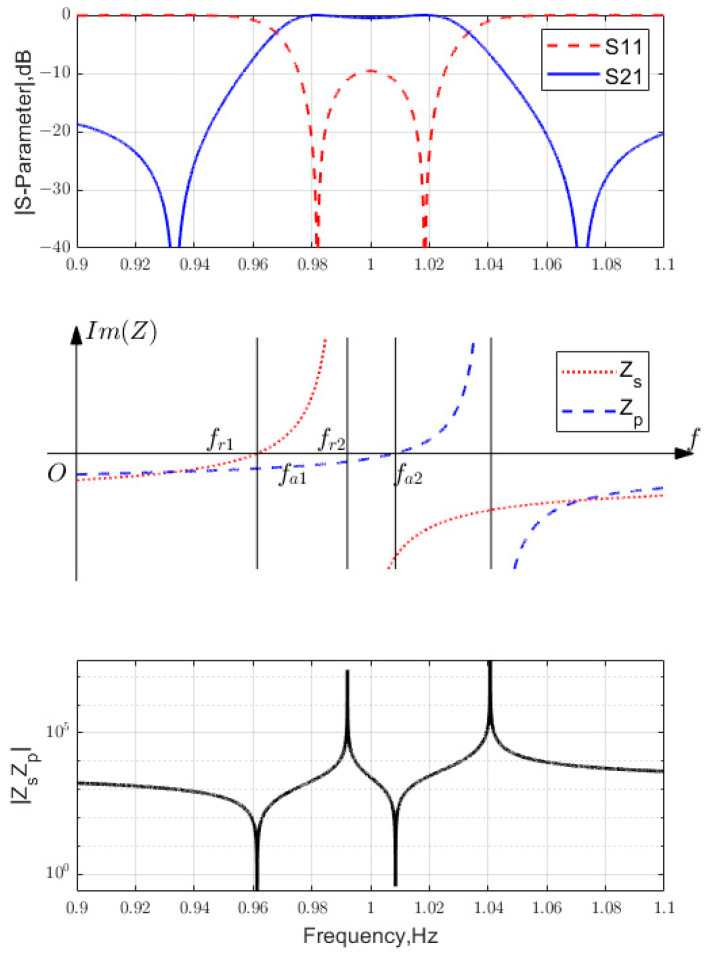
Frequency response of S-parameter and the imaginary part of the acoustic resonator impedance (variable is ${\vec{x}}_{2}$).

**Figure 7 micromachines-15-01075-f007:**
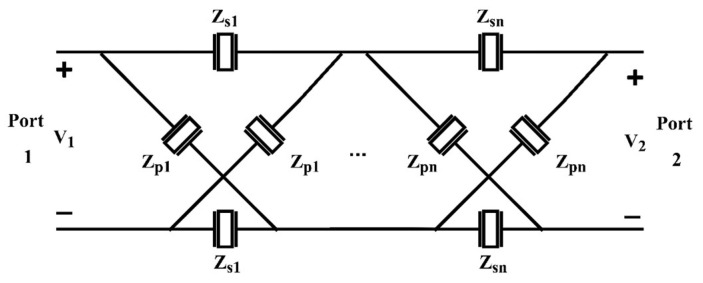
Topology of a multi-stage lattice network.

**Figure 8 micromachines-15-01075-f008:**
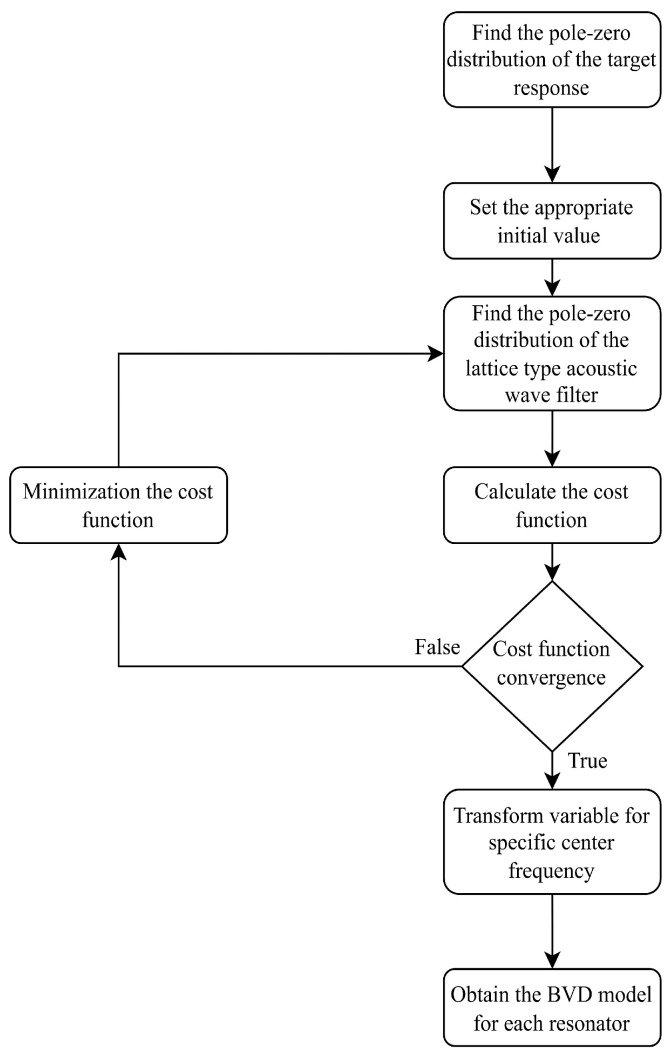
Flow chart for pole-zero distribution optimization.

**Figure 9 micromachines-15-01075-f009:**
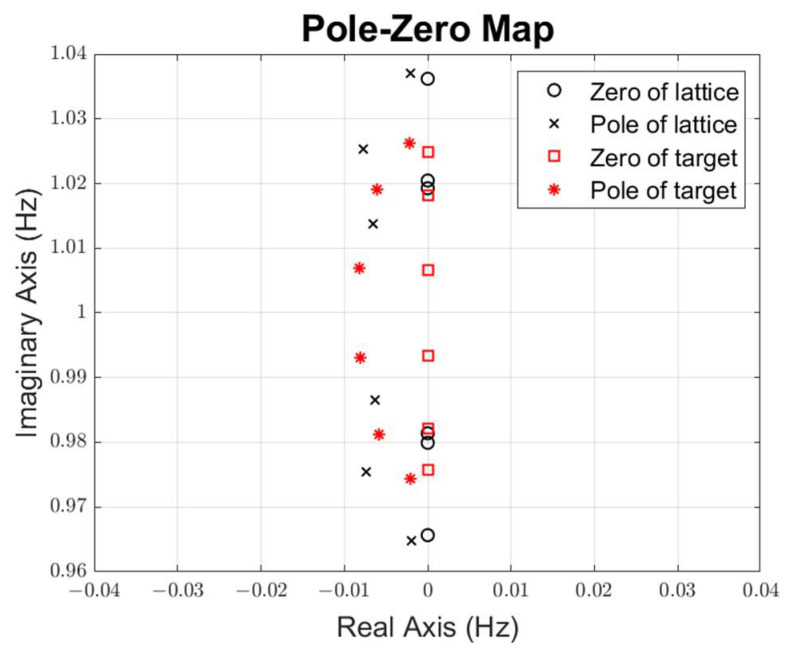
Initial pole-zero distribution.

**Figure 10 micromachines-15-01075-f010:**
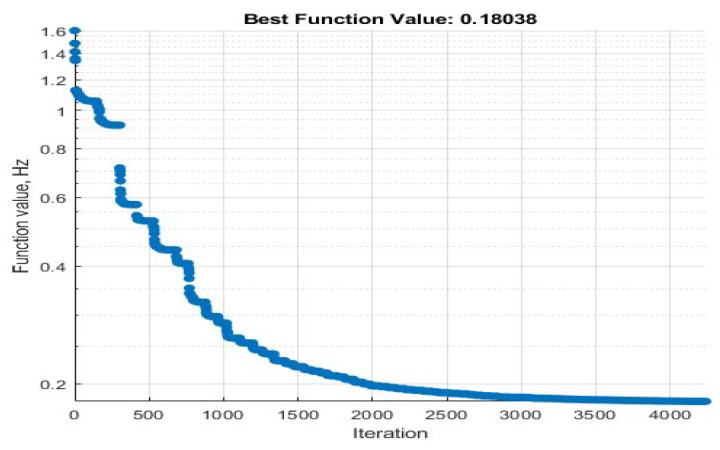
Cost function iteration convergence process.

**Figure 11 micromachines-15-01075-f011:**
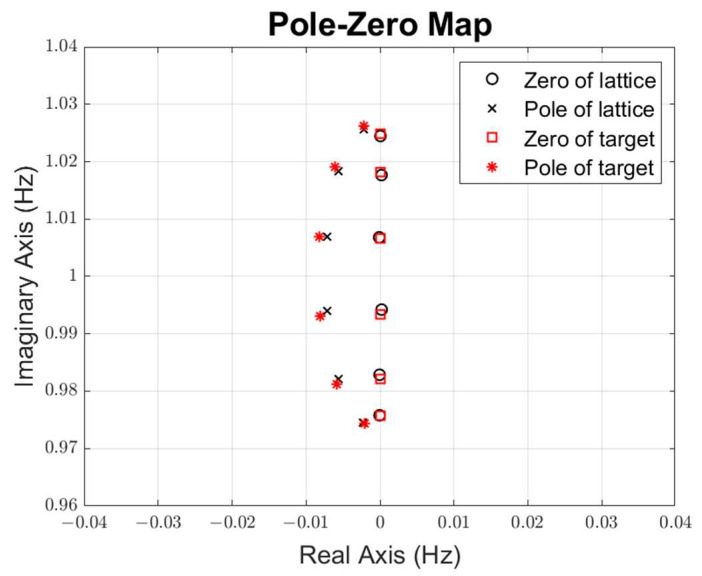
Pole-zero distribution after optimization.

**Figure 12 micromachines-15-01075-f012:**
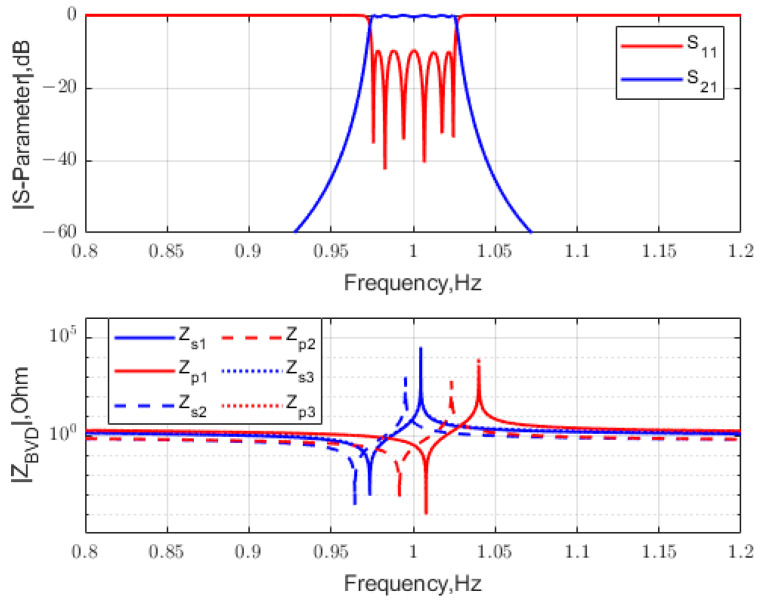
S-parameter after pole-zero distribution optimization for the three-stage lattice filter.

**Figure 13 micromachines-15-01075-f013:**
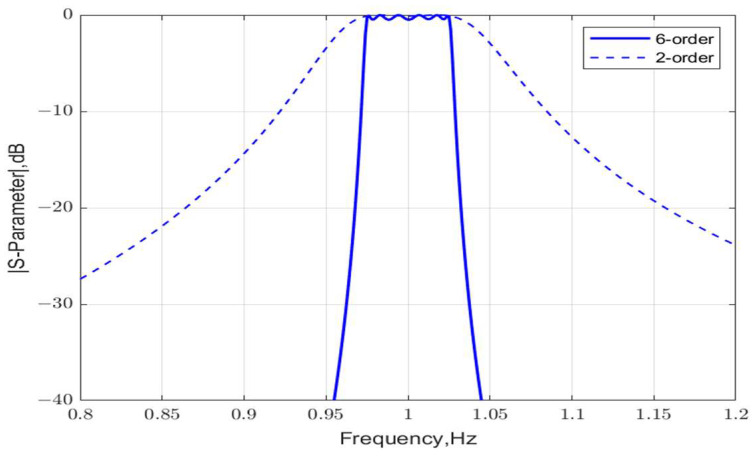
A performance comparison between second-order and sixth-order lattice-type filters with the same bandwidth.

**Figure 14 micromachines-15-01075-f014:**
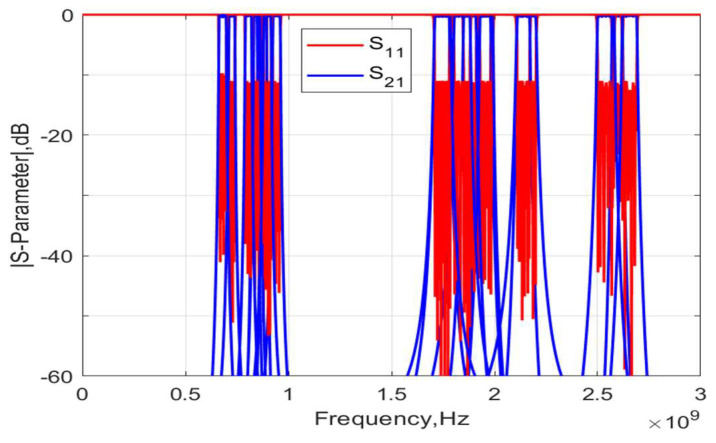
Frequency response of S-parameters for three-stage lattice filters for applications in 5G bands.

**Table 1 micromachines-15-01075-t001:** BVD model parameters of acoustic resonators used in the three-stage lattice filter.

BVD	$Z_{m}\;(\Omega )$	$f_{r}\;(Hz)$	$L_{m}\;(H)$	$C_{m}\;(F)$	$C_{o}\;(F)$
$Z_{s1}$	19.6946	0.9736	3.2196	0.0083	0.1278
$Z_{p1}$	25.8630	1.0079	4.0840	0.0061	0.0940
$Z_{s2}$	10.2497	0.9645	1.6914	0.0161	0.2478
$Z_{p2}$	10.0815	0.9917	1.6180	0.0159	0.2454
$Z_{s3}$	21.8005	0.9736	3.5638	0.0075	0.1154
$Z_{p3}$	25.6740	1.0078	4.0546	0.0062	0.0947

**Table 2 micromachines-15-01075-t002:** The various bands that are required for 5G systems.

5GNR FR1 Band	Uplink Operating Band User Equipment Transmit (MHz)	FBW of Uplink Operating Band	Downlink Operating Band User Equipment Receive (MHz)	FBW of Downlink Operating Band	Duplex Mode	Band Width (MHz)
n1	1920–1980	3.1%	2110–2170	2.8%	FDD	60
n2	1850–1910	3.2%	1930–1990	3.1%	FDD	60
n3	1805–1880	4.1%	1710–1785	4.3%	FDD	75
n5	869–894	2.8%	824–849	2.9%	FDD	25
n7	2620–2690	2.6%	2500–2579	3.1%	FDD	70
n8	925–960	3.7%	880–915	3.9%	FDD	35
n20	832–862	3.5%	791–821	3.7%	FDD	20
n28	703–748	6.2%	758–803	5.8%	FDD	45
n38	2570–2620	1.9%	2570–2620	1.9%	TDD	50
n41	2496–2690	7.5%	2496–2690	7.5%	FDD	194
n66	1710–1780	4.0%	2110–2200	4.2%	FDD	70/90
n71	663–698	5.1%	617–652	5.5%	FDD	35
n77	3300–4200	24%	330–4200	24%	TDD	900
n78	3300–3800	14%	3300–3800	14%	TDD	500
n79	4400–5000	12.7%	4400–5000	12.7%	TDD	600
n81	880–915	3.9%	N/A		SUL	35
n83	703–748	6.2%	N/A		SUL	45

## Data Availability

The original contributions presented in the study are included in the article, further inquiries can be directed to the corresponding author.

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
