# Peer review of "Parametric Synthesis of Single-Stage Lattice-Type Acoustic Wave Filters and Extended Multi-Stage Design"

_micromachines, 2024, doi:10.3390/mi15091075_

Round 1

Reviewer 1 Report

Comments and Suggestions for Authors

This paper discusses the relationships that need to be satisfied between the impedance of series and shunt resonators when building single-stage lattice-type acoustic filters. However, there are infinite kinds of solution results for one equation with two variables, and the paper does not quantitatively analyses the relationship between different solutions and the filter bandwidth and RL.

If only to show that for the lattice topology, a passband takes shape when one branch behaves inductively and the other behaves capacitively, which has been illustrated in previous studies (Q. Yang, W. Pang, D. Zhang and H. Zhang, " ang, Q.; Pang, W.; Zhang, D.; Zhang, H. A Modified Lattice Configuration Design for Compact Wideband Bulk Acoustic Wave Filter Applications. Micromachines 2016, 7, 133.)

1. It is hard to justify the novelty of this manuscript.

2. Lines 108 and 109: Cm = 11.761 and 11.6342 fF should be changed to 1.1761 and 1.16342 pF; C0 = 181.05 and 179.21 fF should be changed to 18.105 and 17.921 pF;

3. Fig. 3, Fig. 4 and Fig. 11: it is difficult to see the flatness in the passband of the filter, please shorten the range of the horizontal coordinates.

4. The last figure is wrongly labelled with the serial number.

5. Multi-stage lattice filter is designed to solve the problem that single-stage filter selectivity and out-of-band rejection are not enough, and it is suggested to add graphs to illustrate the degree of optimization.

6. the symbols in the text of the subscript ‘s / p’ have two meanings: series (resonance) / parallel (anti-resonance), it is recommended to make a distinction.

Comments on the Quality of English Language

Minor editing of English language required.

Author Response

Description of Revisions in Response to the Reviewers’ Comments

We are indebted to the reviewers for the valuable comments. Accordingly, we have made revisions marked in red. The specific revisions are described as below.

Reviewer 1

Comments and Suggestions for Authors

This paper discusses the relationships that need to be satisfied between the impedance of series and shunt resonators when building single-stage lattice-type acoustic filters. However, there are infinite kinds of solution results for one equation with two variables, and the paper does not quantitatively analyses the relationship between different solutions and the filter bandwidth and RL.

Response:

We agree that there is more than one solution for the single-stage lattice filter design. Different solution results can achieve different filter performance. This study does not intend to try all possible solutions, but rather points out two design approaches for different filter performance, one with a narrow passband, and the other with wider passband. The design method for narrow passband is described in subsection 2.3, where the design formulae are derived. We believe this is for the first time the design formulae have been given. The design for single-stage filter with a wider passband is given in subsection 2.4.

Actions:

Please find the first sentence of the Abstract. We emphasize that this study proposes a single-stage lattice-type acoustic filter for either a narrow passband filter or a wider passband filter using two kinds of parameter assignments in Butterworth-Van Dyke (BVD) model.

If only to show that for the lattice topology, a passband takes shape when one branch behaves inductively and the other behaves capacitively, which has been illustrated in previous studies (Q. Yang, W. Pang, D. Zhang and H. Zhang, Modified Lattice Configuration Design for Compact Wideband Bulk Acoustic Wave Filter Applications. Micromachines 2016, 7, 133.)

Response:

We appreciate this expert comment and the cited reference. The design approach for a wider passband is given when one branch behaves inductively and the other capacitively. We agree that the statement has been given by the reference. However, a deeper insight for the relation of parameters is obtained in (11), i.e., . To our knowledge, this is for the first time it has been characterized. The derivation is found in subsection 2.4.  

Actions:

We include the reference mentioned by the Reviewer in this article to improve its coverage. Please find [9] in the revised manuscript. In addition, we add in the Section 1 that
“A modified lattice configuration comprising auxiliary shunt inductors is proposed to achieve a wideband filter response using AlN-based BAW resonators [9].” It is also commented “Although it exhibits the larger bandwidth, its insertion loss is degraded as a tradeoff.” Please find the last two sentences in the fourth paragraph of Section 1.

  1. It is hard to justify the novelty of this manuscript.

Response:

We consider that this study still has novelty because it for the first time,

  • derive the analytic design for single-stage lattice type filters with a narrow passband (in subsection 2.3).
  • derive the analytic relationship of the parameters in BVD model for single-stage lattice type filters with a wider passband (in subsection 2.4)
  • propose a systematic optimization approach for the design parameters in multi-stage lattice type filters to achieve wider passbands and better selectivity (in subsection 2.5).

The main contribution of this study is the ability to quickly obtain the design parameters for a single-stage lattice-type acoustic wave filter based on FBW and RL specifications. These parameters can serve as initial values for the multi-stage lattice-type filter, and the optimization method of pole-zero distribution can be used to speed up the design.

  1. Lines 108 and 109: Cm = 11.761 and 11.6342 fF should be changed to 1.1761 and 1.16342 pF; C0 = 181.05 and 179.21 fF should be changed to 18.105 and 17.921 pF;

Response:

Thanks for pointing out the typo. In fact, it is the inductance unit that needs to be changed. The values Lm = 5.4394 and 5.3842nH should be changed to 543.94 and 538.42 nH. They have been corrected in the revised manuscript.

  1. 3, Fig. 4 and Fig. 11: it is difficult to see the flatness in the passband of the filter, please shorten the range of the horizontal coordinates.

Response:

Thanks for the suggestions. We have shortened the range of the x-coordinate for the three figures in the revised manuscript.

  1. The last figure is wrongly labelled with the serial number.

Response:

Thanks for pointing out the typos. It has been corrected to Fig. 14 in the revised manuscript.

  1. Multi-stage lattice filter is designed to solve the problem that single-stage filter selectivity and out-of-band rejection are not enough, and it is suggested to add graphs to illustrate the degree of optimization.

Response

Thanks for the comments. The multi-stage lattice filter is indeed designed to address the issues of selectivity and out-of-band rejection in single-stage filters. We have included a new graph to illustrate the degree of optimization. It shows the performance comparison between 2nd-order and 6th-order lattice-type filters with the same bandwidth. It is verified that the multi-stage lattice filter design greatly improves the selectivity and out-of-band rejection.

Action:

Please find the last two sentences in the last paragraph of subsection 2.7 and the newly added Fig. 13 in the revised manuscript.

  1. the symbols in the text of the subscript ‘s / p’ have two meanings: series (resonance) / parallel (anti-resonance), it is recommended to make a distinction.

Response:

Thanks for the constructive suggestion. We have changed the subscripts to use 'r' for resonance and 'a' for anti-resonance in the revised manuscript to clearly distinguish between s and p (series and parallel). Note that the subscripts appearing in Fig. 5, Fig. 6, and Table.1 have been corrected accordingly.

Reviewer 2 Report

Comments and Suggestions for Authors

This study is for the parametric synthesis of single-stage lattice-type acoustic wave filters and extended multi-stage design. The contents are interesting and the manuscript is well-written. But it needs some minor revisions to publish in the journal. The comments are as follows:

1. In the Fig 10. it shows the plot after optimization. It looks somewhat better fit than before, but readers cannot see the how was the error between lattice and the target. Please add the error values to the manuscript.

2. What is the BVD model? Please add the description for this to the manuscript for the readers.

3. Does the author have the target commercial product for the single-stage lattice filter? If so, it would be good to add a brief photograph of the commercial product to enhance the understanding of readers.

Author Response

Description of Revisions in Response to the Reviewers’ Comments

We are indebted to the reviewers for the valuable comments. Accordingly, we have made revisions marked in red. The specific revisions are described as below.

Reviewer 2

Comments and Suggestions for Authors

This study is for the parametric synthesis of single-stage lattice-type acoustic wave filters and extended multi-stage design. The contents are interesting and the manuscript is well-written. But it needs some minor revisions to publish in the journal. The comments are as follows:

Response:

Thanks for the comments. We have made the following revisions as suggested.

  1. In the Fig 10. it shows the plot after optimization. It looks somewhat better fit than before, but readers cannot see the how was the error between lattice and the target. Please add the error values to the manuscript.

Response

We have included a new graph in Fig. 10 in the revised manuscript, illustrating error values versus iterations.

  1. What is the BVD model? Please add the description for this to the manuscript for the readers.

Response

Thanks for the comment. The BVD represents Butterworth-Van Dyke. To simplify the analysis in this study, we consider lossless BVD model, although the same approach can also be applied to modified BVD models. We appreciate your reminder that suitable reference should be included. Please find the added reference [11] in the revised manuscript.

  1. Does the author have the target commercial product for the single-stage lattice filter? If so, it would be good to add a brief photograph of the commercial product to enhance the understanding of readers.

Response

The single-stage lattice filters considered in this study can realize a second-order Chebyshev filter with a narrow passband, but the selectivity is not good enough. We do not have target commercial products for this part. In practice, most commercial products are implemented with multi-stage ladder type filters.

Round 2

Reviewer 1 Report

Comments and Suggestions for Authors

The design equations for lattice-type acoustic filters proposed in this study can be generally applied to the design of acoustic filters. In the subsequent research, if some qualification can be given to the solution result for the infinite number of equation solutions, it can be better applied to the practical design.